# Molecular identification of Hymenopteran insects collected by using Malaise traps from Hazarganji Chiltan National Park Quetta, Pakistan

Abid Hussain[1], Asmatullah Kakar[1]*, Mahrukh Naseem[1], Kashif Kamran[1], Zafar Ullah[2], Shehla Shehla[3], Muhammad Kashif Obaid[4]*, Nazeer Ahmed[5], Qaiser Khan[1], Iram Liaqat[6]

1 Department of Zoology, Faculty of Life Sciences, University of Balochistan, Balochistan, Pakistan, 2 Department of Zoology, University of Loralai, Balochistan, Pakistan, 3 Department of Zoology, Abdul Wali Khan University Mardan, Khyber Pakhtunkhwa, Pakistan, 4 State Key Laboratory for Animal Disease Control and Prevention, Key Laboratory of Veterinary Parasitology of Gansu Province, Lanzhou Veterinary Research Institute, Chinese Academy of Agricultural Sciences, Lanzhou, Gansu, China, 5 Balochistan University of Information Technology, Engineering and Management Sciences, Balochistan, Pakistan, 6 Department of Zoology, Government College University, Lahore, Punjab, Pakistan

* kashifobaidkanz@gmail.com (MKO); asmardanzai@yahoo.com (AK)

**Data Availability Statement:** All relevant data are within the manuscript and its Supporting information files.

## Abstract

The order Hymenoptera holds great significance for humans, particularly in tropical and sub-tropical regions, due to its role as a pollinator of wild and cultivated flowering plants, parasites of destructive insects and honey producers. Despite this importance, limited attention has been given to the genetic diversity and molecular identification of Hymenopteran insects in most protected areas. This study provides insights into the first DNA barcode of Hymenopteran insects collected from Hazarganji Chiltan National Park (HCNP) and contributes to the global reference library of DNA barcodes. A total of 784 insect specimens were collected using Malaise traps, out of which 538 (68.62%) specimens were morphologically identified as Hymenopteran insects. The highest abundance of species of Hymenoptera (133/538, 24.72%) was observed during August and least in November (16/538, 2.97%). Genomic DNA extraction was performed individually from 90/538 (16.73%) morphologically identified specimens using the standard phenol-chloroform method, which were subjected separately to the PCR for their molecular confirmation via the amplification of cytochrome c oxidase subunit 1 (*cox1*) gene. The BLAST analyses of obtained sequences showed 91.64% to 100% identities with related sequences and clustered phylogenetically with their corresponding sequences that were reported from Australia, Bulgaria, Canada, Finland, Germany, India, Israel, and Pakistan. Additionally, total of 13 barcode index numbers (BINs) were assigned by Barcode of Life Data Systems (BOLD), out of which 12 were un-unique and one was unique (BOLD: AEU1239) which was assigned for *Anthidium punctatum*. This indicates the potential geographical variation of Hymenopteran population in HCNP. Further comprehensive studies are needed to molecularly confirm the existing insect species in HCNP and evaluate their impacts on the environment, both as beneficial (for example,

**Funding:** The author(s) received no specific funding for this work.

**Competing interests:** The authors have declared that no competing interests exist.

pollination, honey producers and natural enemies) and detrimental (for example, venomous stings, crop damage, and pathogens transmission).

## Introduction

In phylum Arthropoda, the dominant members belong to class Insecta, which comprises numerous insect species from orders such as Coleoptera, Diptera, Hymenoptera and Lepidoptera, all of which hold medical and agricultural interests [1]. Globally, only 20% of insect species have been fully described and named, leaving the majority of species unidentified [2]. These families constitute a major component in terrestrial metazoan biodiversity playing a crucial role in preserving ecological services [3], climate change monitoring [4] and providing beneficial ecosystem services to humans [5, 6]. However, Pakistan's insect diversity has received minimal taxonomic attention due to a lack of taxonomic experts and subsequent lack of insect species descriptions [7, 8]. Additionally, the existing lack of evidence regarding population declines in the face of serious environmental issues such as overgrazing, deforestation, soil erosion and waterlogging [9].

Hymenoptera is the second most diverse order of Insects with over 150,000 described species of ants, wasps, bees and many others [10, 11]. The members of this order carry out the process of pollination and thus play a very significant role in maintaining the structure and function of the forest ecosystem [12–14]. They also exert significant influence over the characteristics of modern terrestrial environments [14, 15]. For example, Hymenoptera species display a broad spectrum of social behavior ranging from solitary lifestyles of parasitic wasps to complex nest networks of super-colonial wood ants and bumble bee family systems [15]. In addition, certain phytophagous hymenopterans can be beneficial to their host plants [12, 16]. These features make Hymenoptera an ideal order for understanding of evolutionary dynamics and cohesion of complex social groups of taxa.

Malaise traps have been widely used to assess the abundance and composition of various insect taxa specifically Hymenoptera and Diptera [17–19]. This type of traps consists large netting tent often made out of a fine mesh material, which are advantageous due to low maintenance requirements. Further, these traps provide a comprehensive snapshot of the local insect community [20–22] and several studies have highlighted their effectiveness in capturing Hymenopteran insects [23, 24].

Traditional morphological based approach to insect taxonomy has long been established as an effective means of specie description and identification [17]. However, these methods are challenged [18, 19, 25] by a variety of juvenile life stages of insects, individuals with fluctuating phenotypes and cryptic species that complicate distinct identification [18, 26–30]. To address this issue, the use of molecular techniques specifically DNA Barcoding have been employed to enable large-scale assessments of biodiversity [29–31]. The mitochondrial cytochrome c oxidase subunit 1 (*cox1*) gene is most commonly used as a genetic marker due to the presence of conserved region [26, 32] allowing for discrimination between various groups or biotypes or 'cryptic species' in a single species [33, 34]. The use of Barcode Index Numbers (BINs) in these studies boundaries by comprehensively presenting species diversity [35, 36]. In the order Hymenoptera, DNA barcoding evaluations have reported minor variation at the species level and demonstrated its ability to accurately identify different species [27, 37, 38].

Several studies on morpho-molecular characterization of faunal Hymenopteran fauna in Pakistan have been documented [39–45]. However, the exact distribution and dispersion of

Hymenopteran insects in the Balochistan province remains poorly undocumented due to lack of comprehensive research on the subject. Morphology-based studies on Hymenopteran species in Balochistan have identified specific species such as *Polistes gallicus* (Linneus, 1767) and *Vespula germanica* (Fabricius, 1793) in district Quetta [46]. In district Killa Saifullah of Balochistan, two studies reported nine species including *Polistes flavus* (Cresson, 1868), *Polistes greeniptica* (Fabricius, 1804), *Polistes wattii* (Cameron 1900), *Polistes olivaceous* (DeGeer, 1773), *Polistes indicus* (Stolfa, 1934), *Polistes stigma* (Fabricius, 1793), *Ropalidia brevata* [47] and *Vespa orientalis* (Linnaeus, 1771) [48]. This study serves to bridge a current knowledge gap by performing the first-ever molecular identification of Hymenopteran insects that have been reported from the protected area (HCNP) of Balochistan. This process has the potential to assist taxonomists to accurately classifying these Hymenopteran insects.

## Methods and materials

### Ethical statement

The Advanced Studies and Research Board committee at the University of Balochistan, Quetta has approved this study under registration number UoB/Reg/GSO/1197.

### Study location and Malaise traps setting

The present study was carried out in Hazarganji-Chiltan National Park (HCNP) (30˚13'21.8"N 66˚44'13.5"E), located 20 kilometers Southwest of Quetta city in the Balochistan province, Pakistan. This national park is renowned for its diverse fauna and flora that has been officially designated as the 25th national park in Pakistan. It falls under the International Union for Conservation of Nature (IUCN) Category-V classified as a protected landscape [49], and covers an area of approximately 1315.22 km$^2$. The study area is located at an elevation of about 5500 feet and experiences an annual total precipitation of around 240 mm during the winter season. The average temperature in summer season (June–August) can reach up to 40˚C, while the winter season (November–March) sees rainfall and snowfall with temperature dropping as low as -12˚C [50].

Malaise traps made up of mash material were used. These traps were provided by the International Barcode of Life (IBOL) and designed especially for the capturing of insects. The geographic coordinates of each collection site were recorded using a global positioning system and data obtained were processed in Microsoft Excel 2013 (Microsoft 365®) to create a study map using ArcGIS v 10.3.1 (Fig 1). Five significant locations within the study area were selected for the placement of Malaise traps to collect the insect specimens. These locations include the sub-campus of Balochistan University of Information Technology and Management Sciences (BUITM) (30˚05'14.6"N 66˚56'01.7"E), Hazarganji Nullah (30˚02'10.9"N 66˚52'02.9"E), Kangari (30˚03'08.9"N 66˚55'02.9"E), Wadd (30˚01'01.6"N 66˚49'32.4"E) and Garak (30˚07'37.36"N 66˚43'33.52"E).

### Insect collection and preservation

Insects were collected using Malaise traps from April 2019 to November 2019 (a total of eight months of sampling). Specimens were collected in 500 mL plastic Nalgene® bottles containing 400 ml ethanol (95%) attached to each Malaise trap and then transferred into Whirl-Pak bag® containing 95% ethanol [51]. Specimen collection was performed on weekends (i.e., Saturday and Sunday). The dates of collection were marked on bags and these specimens were brought to the Entomology Laboratory at the University of Balochistan, Quetta for further molecular analyses.

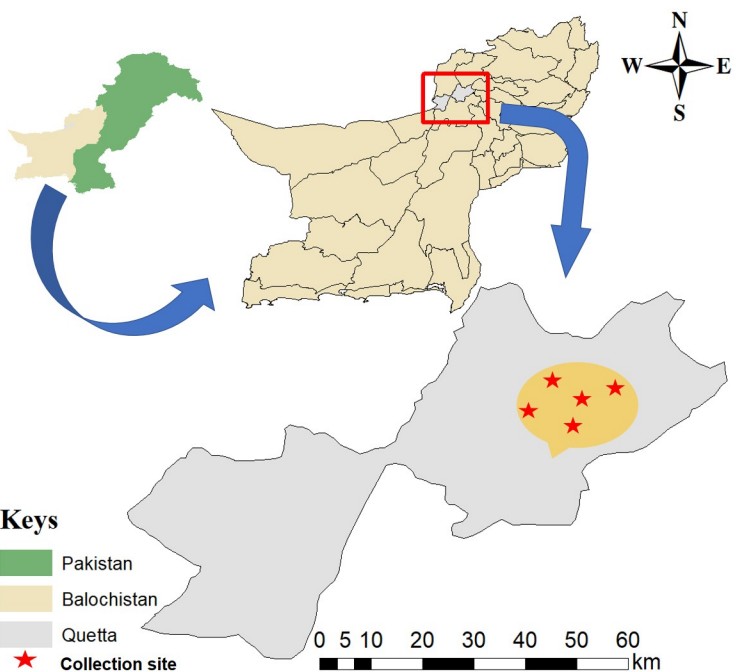

**Fig 1. Map shows the sampling sites of the study area.** This map was created using of software ArcGIS v 10.3.1 (https://www.esri.com/en-us/arcgis/products/arcgis-desktop/resources).

## Morphological identification

The collected specimens underwent a cleaning process using 70% ethanol and rinsed with distilled water to remove any external impurities. A stereomicroscope (Olympus SZ61, Japan) was used to identify the specimens by observing diagnostic features such as color pattern, wing venation, body shape, antennae and head. These characters were examined using established reference materials including standard published keys [52, 53] catalogs and electronic keys [54–56]. In the present study, we follow the family and subfamily classifications [18, 57–59] with additional resolution from the published articles [60, 61]. Identification was also performed by comparison with the help of other available Hymenopteran specimens already identified in the collections of Hymenopteran insects housed at National Insect Museum, National Agricultural Research Centre, Islamabad Pakistan (https://www.parc.gov.pk/).

## Molecular analyses

Genomic DNA extraction was performed from each morphologically identified specimen using the standard phenol-chloroform method [62, 63]. Each specimen was individually homogenized in 200 μl of phosphate-buffered saline (pH = 7.4, 137 mM NaCl, 2.7 mM KCl, 8 mM $Na_2HPO_4$ and 2 mM $KH_2PO_4$). Homogenization was carried out by cutting one or two legs of each insect using sterile scissors and then grinding with a sterile pestle in a 1.5 mL Eppendorf tube. The resulting homogenate was used for DNA extraction. The quality and quantity of the extracted DNA were measured using NanoDrop (NanoQ, Optizen, Daejeon, South Korea) and then stored at -20°C for further analyses. The extracted genomic DNA from each morphologically identified specimen was used for conventional PCR (GE-96G, BIOER, Hangzhou, China) to amplify the universal genetic marker, partial fragments of *cox1* gene (HC02198: 5′-TAA ACT TCA GGG TGA CCA AAA AAT CA-3′ and LCO1490: 5′-

GGT CAA CAA ATC ATA AAG ATA TTG G-3') [64]. The PCR cycling conditions were as follows: initial denaturation at 98 ˚C for 30s, followed by 40 cycles of denaturation at 98 ˚C for 10s, annealing at 63 ˚C for 20s, elongation at 72 ˚C for 25s and a final extension at 72 ˚C for 5 minutes. Each PCR reaction mixture was prepared in 20 μL, consisting of 1 μL of each primer (at a concentration of 10 pmol/μL), 4 μL PCR water, 2 μL (100 ng/μL) genomic DNA and 12 μL Dream*Taq* MasterMix (2X) (Thermo Fisher Scientific, USA). The PCR amplified products were run on a 2% agarose gel prepared in Tris borate EDTA (TBE) containing 2 μl ethidium bromide at a concentration of 0.2–0.5 μg/mL for staining purpose. The resulting bands were observed using Gel Documentation System (BioDoc-It™ Imaging Systems, UVP, LLC).

### Phylogenetic analyses

The obtained amplified products were purified using GeneClean II Kit (Qbiogene, Illkirch, France) following the manufacturer's protocol. The *cox1* partial fragments were subjected to bidirectional sequencing through a commercial Korean company (Macrogen, Inc., Seoul, South Korea). The resulting bidirectional sequences were processed and refined by eliminating the poor reading and contaminated regions using SeqMan (V.5 DNASTAR, USA). The final trimmed and consensus sequences were further analyzed using Basic Local Alignment Search Tool (BLAST) [65] at National Center for Biotechnology Information (NCBI) (https://www.ncbi.nlm.nih.gov/). The sequences along with BINs and other related taxonomic information were recorded on BOLD (https://www.boldsystems.org/) and also deposited in the GenBank (NCBI). Then these sequences with high identity were downloaded in FASTA format from NCBI and were aligned using ClustalW and multiple alignments [66] that were further analyzed using BioEdit alignment editor tool (V.7.0.5, Raleigh, NC, USA) [67]. The Neighbor-Joining method employing the Kimura 2-parameter model was applied and 1000 bootstrap replicates were generated using Molecular Evolutionary Genetic Analysis (MEGA-X) software to construct the phylogenetic tree [68].

## Results

### Morphological identification

A total of 784 insects were captured using Malaise traps in the study area. The highest number of insects was collected at the BUITM sub-campus in Hazar Ganji (177/784, 22.58%), followed by Kangari (162/784, 20.66%), Hazarganji Nullah (154/784, 19.64%), Garak, (149/784, 19.01%) and Wadd (142/784, 18.11%). A total of 538/784 (68.62%) collected specimens were identified as Hymenopteran insects (S1 Fig). The most commonly identified Hymenopteran insects were Bethylidae sp. (59/538, 9.11%), followed by *Tachysphex incertus* (Radoszkowski, 1877) (48/538, 8.92%), *Cerceris rybyensis* (Linnaeus, 1771) (36/538, 8.55%), *Tachytes freygessneri* (43/538, 7.99%), Formicidae sp. and *Lasioglossum* sp. (42/538, 7.81%), Hymenoptera sp. (41/538, 7.62%), *Anthidium punctatum*, *Camponotus compressus* and *Tachysphex* sp. (39/538, 7.25%), *Megachile leachella* (38/538, 7.06%), Sphecidae sp. (37/538, 6.88%) and Evaniidae sp. (35/538, 6.51%) as presented in Table 1.

### Seasonal distribution

Out of 784 captured Hymenopteran insects, only 538 (68.62%) morphologically identified insects were reported on monthly basis. The highest number (133/538, 24.72%) of insects were reported in August, followed by the second highest count in July (114/538, 21.19%), September 93/538 (17.29%), June 68/538 (12.64%), May 57/538 (10.59%), April 31/538 (5.76%), October 26/538 (4.83%), while November had the least number of insects (16/538, 2.97%). This data

**Table 1. Collection sites, number of insects collected and their morpho-molecular characterization.**

| Collection sites | No. of insects collected | Morphologically Identified species | Number | Molecular Characterization | Sequences |
|---|---|---|---|---|---|
| **BUITM sub-campus** | 177 | *Tachysphex incertus* | 10 | 2 | 2 |
| | | *Tachytes freygessneri* | 9 | 2 | 2 |
| | | *Anthidium punctatum* | 8 | 2 | 2 |
| | | *Megachile leachella* | 9 | 1 | 1 |
| | | *Cerceris rybyensis* | 9 | 2 | 2 |
| | | *Camponotus compressus* | 8 | 1 | 1 |
| | | *Lasioglossum* sp. | 9 | 1 | 1 |
| | | *Tachysphex* sp. | 10 | 2 | 2 |
| | | Hymenoptera sp. | 8 | 1 | 1 |
| | | Evaniidae sp. | 7 | 2 | 2 |
| | | Formicidae sp. | 10 | 1 | 1 |
| | | Sphecidae sp. | 8 | 1 | 1 |
| | | Bethylidae sp. | 12 | 2 | 2 |
| **Hazarganji Nullah** | 154 | *Tachysphex incertus* | 11 | 1 | 1 |
| | | *Tachytes freygessneri* | 7 | 1 | 1 |
| | | *Anthidium punctatum* | 9 | 2 | 2 |
| | | *Megachile leachella* | 8 | 1 | 1 |
| | | *Cerceris rybyensis* | 10 | 2 | 2 |
| | | *Camponotus compressus* | 7 | 1 | 1 |
| | | *Lasioglossum* sp. | 8 | 1 | 1 |
| | | *Tachysphex* sp. | 5 | 1 | 1 |
| | | Hymenoptera sp. | 8 | 2 | 2 |
| | | Evaniidae sp. | 7 | 2 | 2 |
| | | Formicidae sp. | 7 | 1 | 1 |
| | | Sphecidae sp. | 6 | 1 | 1 |
| | | Bethylidae sp. | 9 | 1 | 1 |
| **Kangari** | 162 | *Tachysphex incertus* | 8 | 2 | 2 |
| | | *Tachytes freygessneri* | 10 | 1 | 1 |
| | | *Anthidium punctatum* | 6 | 2 | 2 |
| | | *Megachile leachella* | 7 | 1 | 1 |
| | | *Cerceris rybyensis* | 8 | 2 | 2 |
| | | *Camponotus compressus* | 6 | 1 | 1 |
| | | *Lasioglossum* sp. | 9 | 1 | 1 |
| | | *Tachysphex* sp. | 8 | 1 | 1 |
| | | Hymenoptera sp. | 9 | 1 | 1 |
| | | Evaniidae sp. | 7 | 2 | 2 |
| | | Formicidae sp. | 9 | 1 | 1 |
| | | Sphecidae sp. | 10 | 1 | 1 |
| | | Bethylidae sp. | 11 | 2 | 2 |

(*Continued*)

**Table 1.** (Continued)

| Collection sites | No. of insects collected | Morphologically Identified species | Number | Molecular Characterization | Sequences |
|---|---|---|---|---|---|
| **Wadd** | 142 | *Tachysphex incertus* | 10 | 2 | 2 |
| | | *Tachytes freygessneri* | 9 | 1 | 1 |
| | | *Anthidium punctatum* | 7 | 1 | 1 |
| | | *Megachile leachella* | 8 | 2 | 2 |
| | | *Cerceris rybyensis* | 9 | 1 | 1 |
| | | *Camponotus compressus* | 10 | 1 | 1 |
| | | *Lasioglossum* sp. | 7 | 1 | 1 |
| | | *Tachysphex* sp. | 10 | 2 | 2 |
| | | Hymenoptera sp. | 6 | 1 | 1 |
| | | Evaniidae sp. | 8 | 1 | 1 |
| | | Formicidae sp. | 7 | 1 | 1 |
| | | Sphecidae sp. | 5 | 2 | 2 |
| | | Bethylidae sp. | 8 | 2 | 2 |
| **Garak** | 149 | *Tachysphex incertus* | 9 | 2 | 2 |
| | | *Tachytes freygessneri* | 8 | 1 | 1 |
| | | *Anthidium punctatum* | 9 | 1 | 1 |
| | | *Megachile leachella* | 6 | 2 | 2 |
| | | *Cerceris rybyensis* | 10 | 2 | 2 |
| | | *Camponotus compressus* | 8 | 1 | 1 |
| | | *Lasioglossum* sp. | 9 | 1 | 1 |
| | | *Tachysphex* sp. | 6 | 1 | 1 |
| | | Hymenoptera sp. | 10 | 1 | 1 |
| | | Evaniidae sp. | 6 | 2 | 2 |
| | | Formicidae sp. | 9 | 1 | 1 |
| | | Sphecidae sp. | 8 | 1 | 1 |
| | | Bethylidae sp. | 9 | 1 | 1 |
| **Total collection** | 784 | *Tachysphex incertus* | 48 | 9 | 9 |
| | | *Tachytes freygessneri* | 43 | 6 | 6 |
| | | *Anthidium punctatum* | 39 | 8 | 8 |
| | | *Megachile leachella* | 38 | 7 | 7 |
| | | *Cerceris rybyensis* | 36 | 9 | 9 |
| | | *Camponotus compressus* | 39 | 5 | 5 |
| | | *Lasioglossum* sp. | 42 | 5 | 5 |
| | | *Tachysphex* sp. | 39 | 7 | 7 |
| | | Hymenoptera sp. | 41 | 6 | 6 |
| | | Evaniidae sp. | 35 | 9 | 9 |
| | | Formicidae sp. | 42 | 5 | 5 |
| | | Sphecidae sp. | 37 | 6 | 6 |
| | | Bethylidae sp. | 59 | 8 | 8 |
| **Total** | | | **538** | **90** | **90** |

indicates that the distribution of Hymenopteran insects follows a Gaussian distribution pattern (Fig 2).

## Molecular confirmation

A total of 90 (16.73%) out of 538 (one or two specimens of morphologically identified Hymenopteran insects from each study location) were used for molecular confirmation. The

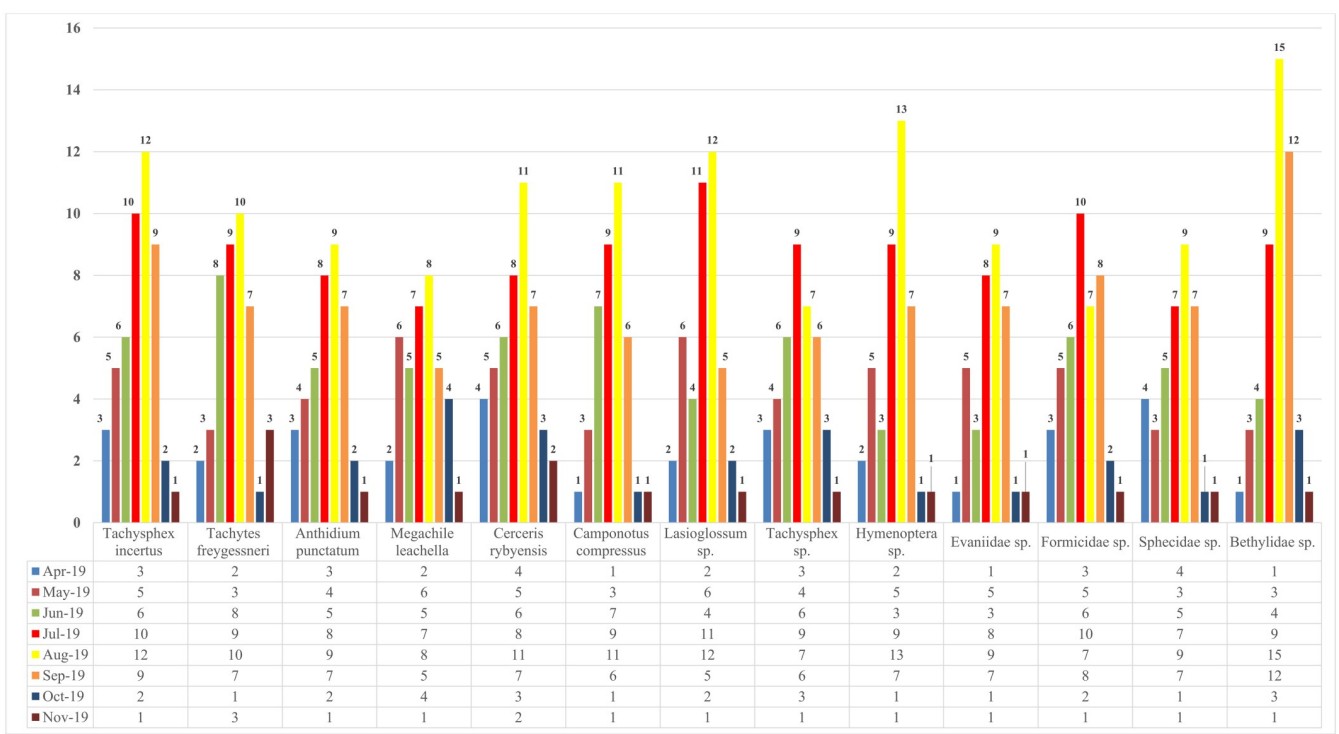

**Fig 2. Month-wise (April 2019 to November 2019) distribution of Hymenopteran insects.**

remainder of the insect species in the Malaise traps belonged to other insect orders (unpublished data, manuscript under preparation). Of these, six insect species were identified at specie level, namely *Tachysphex incertus* (Radoszkowski, 1877) (n = 9), *Cerceris rybyensis* (Linnaeus, 1771) (n = 9), *Anthidium punctatum* (Latreille, 1809) (n = 8), *Megachile leachella* (Curtis 1828) (n = 7), *Tachytes freygessneri* (Kohl, 1881) (n = 6), *Camponotus compressus* (Fabricius 1787) (n = 5). Two insects were identified at genera level including *Tachysphex* sp. (n = 7), and *Lasioglossum* sp. (n = 5). Four different Hymenopteran insects were identified at family level including Evaniidae sp. (n = 9), Bethylidae sp. (n = 8), Sphecidae sp. (n = 6) and Formicidae sp. (n = 5), while one group of insects was identified at the level of order as Hymenoptera sp. (n = 6) as shown in Table 1.

## Phylogenetic analysis outputs

A phylogenetic tree was constructed using the Neighbor-Joining (NJ) method for 13 sequences (Fig 3). The obtained identical sequences were treated as consensus sequences for each Hymenopteran insect. The BLAST analysis revealed sequence similarities ranging from 91.64% to 100% which provided strong support for the reliability of phylogenetic tree fidelity. For example, *Tachysphex incertus* (658 bp) exhibited a 98.48% identity with *Tachysphex incertus*, *Tachytes freygessneri* (658 bp) showed 99.54% similarity with *Tachytes freygessneri*, *Anthidium punctatum* (614 bp) had 92.83%-93.16% identity with *Anthidium punctatum*, *Megachile leachella* (539 bp) showed 100% with *Megachile leachella*, *Cerceris rybyensis* (658 bp) showed 91.64% identity with *Cerceris rybyensis*, *Camponotus compressus* (617 bp) showed 93.37% with *Camponotus compressus*, *Lasioglossum* sp. (637 bp) showed 99.84% to 100% with *Lasioglossum* sp., *Tachysphex* sp. (658 bp) showed 93.92% with *Tachysphex* sp., Hymenoptera sp. (603 bp)

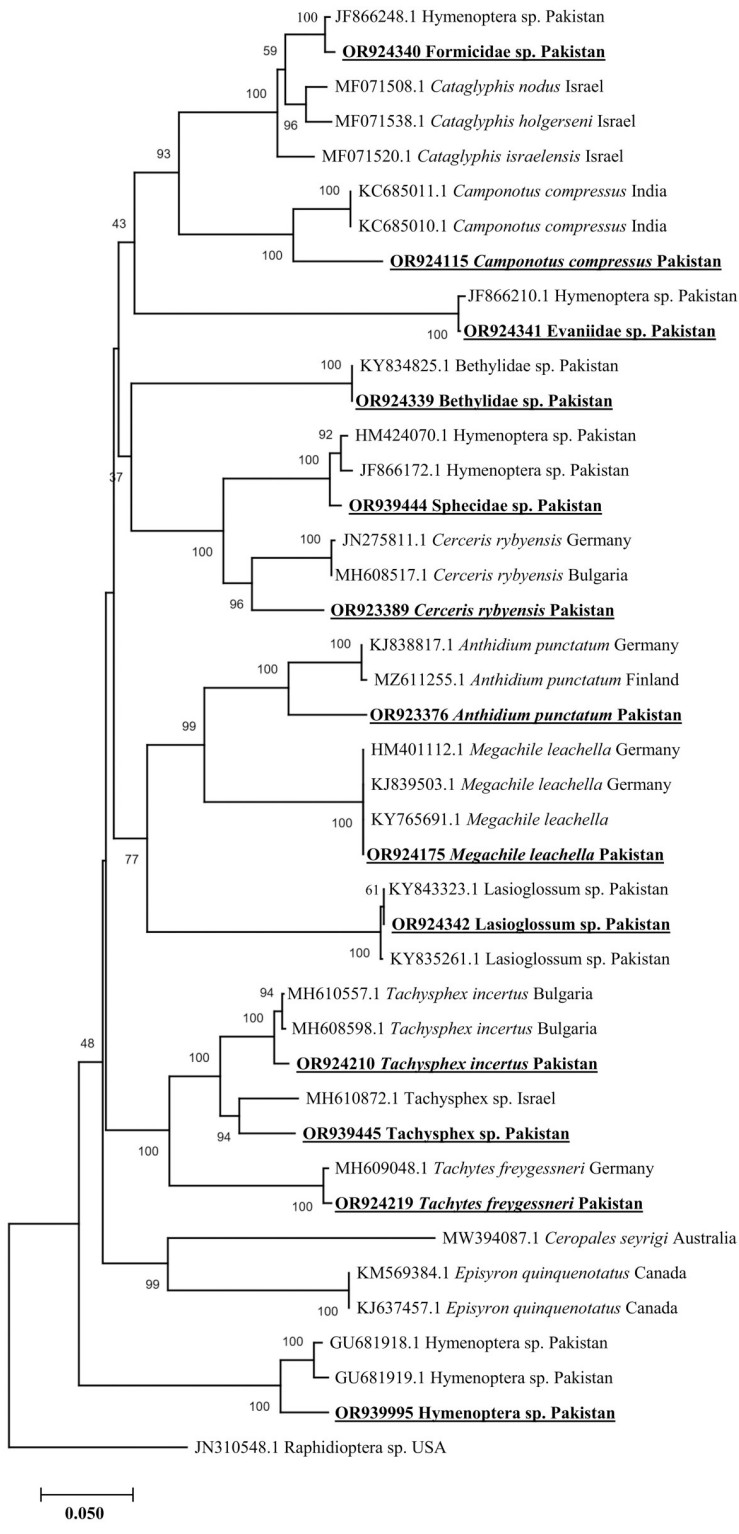

**Fig 3. For order Hymenoptera, the phylogenetic tree based on *cox1* partial fragments sequences was constructed via the Neighbor-Joining method with the Kimura 2-parameter model.** The bootstrap values (1000 replications) are shown at each node. Raphidioptera sp. (JN310548.1) from USA was selected as an outgroup. The obtained sequences of this study are marked in bold and also underlined.

**Table 2. Identification of Hymenoptera insect species through DNA barcoding method and their close matching results are presented.**

| Specimen code | Species | Own BINS | Nearest BINS | Accession number (NCBI) |
|---|---|---|---|---|
| AQBL-2 | *Tachytes freygessneri* (Kohl, 1881) | BOLD:ACH384 | BOLD:ADM7708 | OR924219 |
| AQBL-7 | *Anthidium punctatum* (Latreille, 1809) | BOLD:AEU1239 | BOLD:AEH2201 | OR923376 |
| AQBL-8 | *Cerceris rybyensis* (Linnaeus,1771) | BOLD:AEJ5559 | BOLD:ACY9461 | OR923389 |
| AQBL-9 | *Tachysphex incertus* (Radoszkowski, 1877) | BOLD:AAV7059 | BOLD:ACX5234 | OR924210 |
| AQBL-10 | *Tachysphex* sp. ∗ | BOLD:AET4720 | BOLD:ACZ0155 | OR939445 |
| AQBL-16 | Evaniidae sp. ‡ | BOLD:AAQ0495 | BOLD:ABW3800 | OR924341 |
| AQBL-20 | Sphecidae sp. ‡ | BOLD:AAG8316 | BOLD:ADU2012 | OR939444 |
| AQBL-21 | Bethylidae sp. ‡ | BOLD:ACD9427 | BOLD:AEA2483 | OR924339 |
| AQBL-27 | Hymenoptera sp. ‡∗ | BOLD:AES2580 | BOLD:AAG8309 | OR939995 |
| AQBL-28 | *Lasioglossum* sp. ∗ | BOLD:ACA2731 | BOLD:ADJ7874 | OR924342 |
| AQBL-38 | *Megachile leachella* (Curtis, 1828) | BOLD:AAD2767 | BOLD:AAN4635 | OR924175 |
| AQBL-45 | Formicidae sp. ‡ | BOLD:AAQ0513 | BOLD:AEK5824 | OR924340 |
| AQBL-41 | *Camponotus compressus* (Fabricius 1787) | BOLD:ADZ9693 | BOLD:AEE5073 | OR924115 |

Symbol used in the above table: (∗) denotes the representation of genera, (‡) signifies the representation of families, while (‡*) represents the order Hymenoptera.

showed 94.53% to 95.02% with Hymenoptera sp., Evaniidae sp. (655 bp) showed 99.48% with Hymenoptera sp., Formicidae sp. (580 bp) showed 99.14% with Hymenoptera sp., Sphecidae sp. (638 bp) showed 98.43% to 98.59% with Hymenoptera sp. and Bethylidae sp. (557 bp) had 100% with Bethylidae sp. These clusters were samples from various nations across the world including Australia, Bulgaria, Canada, Finland, Germany, India, Israel and Pakistan (Fig 3).

## BOLD and GenBank databases

All the details regarding specific identified species along with their specific specimen numbers, images, their BINs, and accession numbers are available in BOLD and GenBank databases as presented in Table 2. The results of the BOLD database analysis showed that 13 BINs were assigned for the obtained consensus sequences with six of these assigned to the species level, two to the genera level, four to the family level, and one to the order level. Of the 12 BINs that had been previously assigned (un-unique), one unique BIN (BOLD: AEU1239) was assigned to *Anthidium punctatum* in the study region, indicating the potential presence of a new species in HCNP, Balochistan.

## Discussion

The current study aimed to expand scientific understating of Hymenoptera diversity in Balochistan given their lack of documentation. Previous faunistic studies relied on morphological analysis of species with medical and economic importance [46, 69–73]. These morphological studies are essential for determining species abundance in a given geographical area and monitoring the long-term shifts in population trends and diversity [73]. The results of these investigations indicates the biodiversity of insects and their importance in Pakistan's protected areas such as HCNP [74]. Though these areas have received minimal research attention notably on Hymenopteran. In order to gain reliable data on Hymenopteran diversity, molecular technique was used in this study for their molecular confirmation.

In the present study, different Hymenopteran insects were molecularly identified, and corresponding BINs were generated, in which one was unique that was indicating the lack of representation in barcode database. In the IBOL database, all species codes and their

morphologically identified images were displayed. DNA barcoding serves as an invaluable tool for large-scale, high-throughput taxonomic classifications offering potential implications for biodiversity investigations in protected regions. Despite this potential capacity, the diversity of Hymenopteran insects in Balochistan has not been adequately explored. For instance, Ashfaq et al. [75] conducted extensive research on identification of dominant insect orders including Coleoptera, Hemiptera, Hymenoptera and Lepidoptera that were captured by using Malaise traps in Pakistan. Studies have demonstrated that the HCNP, Hingol National Park (HNP) and other protected areas in Balochistan serve as biodiversity hotspots for various insects including bees, wasps, ants and sawflies [76]. Limited information regarding different Hymenopteran insects pose significant economic and health risks to local population of insects in Balochistan [77]. There have been noteworthy contributions of using DNA barcoding to identify various Hymenoptera bee species [78]. Insects in the Himalayas [79], conserved moderate-climate regions in Canada [80], South African savannah termite species [81], insect species in the Amazon jungle [82], New York city community garden bees [83] and insects from Thailand's national parks [84] have been identified via DNA barcoding.

Our findings are consistent with previous investigations, which have used a combination of basic morphological and DNA barcode approaches [27, 85, 86]. Notably, one study relied mainly on the morphological features and reported almost 70% identification rate of various insects; this figure is closely matched by the 68.62% rate obtained in our findings [86]. The application of molecular identification has been widely accepted as a successful species identification method, solving issues which arise due to morphological similarities and concealed variations within cryptic species [87]. Several studies have used *cox1* gene for the molecular identification of insect species [8, 38, 78]. By conducting phylogenetic analysis of sequences obtained in this study, we determined an evolutionary linkage between various obtained insects and their corresponding species, each showing the percent similarity ranging from 91.64% to 100%. Furthermore, genomes data from Canada, Germany, United States of America, United Kingdom, Pakistan and India have produced similar phylogenetic patterns. The cladding of the obtained sequences of different insects in phylogenetic tree is representing that the obtained insects are similar as in the aforementioned various countries [8, 78, 86, 88–90].

In this study, collected data regarding seasonal variations were analyzed that have impacts on the distribution of Hymenopteran insects. The average temperature from May to September was high and creating an ideal environment for insects to proliferate. However, a decrease in average temperature from September to April resulted in decline of their population. The average population of Hymenopteran insects showed a pattern of Gaussian distribution with highest number recorded in August (24.72%) and the least in November (2.97%). The Gaussian distribution has been described in other studies that were presenting the various Hymenopteran characteristics such as time spent on host feeding, host acceptance, host suitability, time spent walking, and body size [91, 92]. The seasonal variations of Hymenopteran insects can be better understood by studying the influence of environmental factors such as humidity, temperature, precipitation, and food availability [93, 94]. The obtained results can help to develop the mitigation strategies to overcome the substantial economic and health concerns of the local insect population in Pakistan [77]. Other influencing factors such as shortage of food, competition for resources, temperature, snow-fall and habitat destruction can also contribute to the survivorship of insects during the colder months [95, 96].

This study is limited by a few factors such as it's a short-term focus on seasonal collections which may not fully account the yearly fluctuation of Hymenopteran insects' population, thereby potentially obscuring meaningful distribution trends and species with distinct life cycles or behaviors. Moreover, the availability of DNA barcodes in majority of specimens was likely hindered by probable contamination of samples, resulting the degradation of DNA. In

order to gain a more substantial insights into the ecology, behavior and evolution of the Hymenopteran insects in other protected areas, future studies should build upon the foundations established by this study. This study does not present any evidence of cryptic species, though it does emphasize the difficulty of species identification due to the presence of cryptic species and morphological convergence [97].

## Conclusion

Fron the current study, we conclude that the HCNP has a rich of Hymenopteran insects' fauna. DNA barcoding confirmed a total of six different species at their specie level, two at genus level, four at family level, and one at order level. The BOLD database has identified 13 BINs for Hymenopteran insects, of which one unique BIN was obtained for *Anthidium punctatum*. Furthermore, temperature variability throughout the year was found to exhibits a Gaussian distribution pattern. To further validate the molecular identities of insect species, it is recommended to perform further comprehensive studies in order to know about the diversification of insect species in other protected areas of Pakistan.

## Supporting information

**S1 Fig.** Figure presents a taxonomically arranged list of morphologically identified Hymenopterans insects: (A) *Anthidium punctatum* (B) *Bethylidae* sp. (C) *Camponotus compressus* (D) *Ceropales seyrigi* (D) *Evaniidae* sp. (F) Formicidae sp. (G) *Lasioglossum* sp. (H) *Megachile leachella* (I) Hymenoptera sp. (J) Sphecidae sp. (K) *Tachysphex incertus* (L) *Tachysphex* sp. The datasets presented in the study have been deposited to the NCBI GenBank repository (ncbi. nlm.nih.gov), and the obtained accession numbers are OR923389 (*Cerceris rybyensis*), OR924175 (*Megachile leachella*), OR923376 (*Anthidium punctatum*), OR924115 (*Camponotus compressus*), OR924210 (*Tachysphex incertus*), OR924219 (*Tachytes freygessneri*), OR939445 (*Tachysphex* sp.), OR924342 (*Lasioglossum* sp.), OR924339 (Bethylidae sp.), OR924340 (Formicidae sp.), OR924341 (Evaniidae sp.), OR939444 (Sphecidae sp.), and OR939995 (Hymenoptera sp.).
(TIF)

## Acknowledgments

Authors are thankful to the workers at HCNP who assist us in collecting of Hymenopteran fauna. We are thankful to Prof. Dr. Muhammad Ashfaq Ahmed (University of Balochistan) for the editing the revised manuscript.

## Author Contributions

**Conceptualization:** Asmatullah Kakar, Mahrukh Naseem.

**Data curation:** Abid Hussain, Kashif Kamran, Zafar Ullah, Shehla Shehla, Muhammad Kashif Obaid.

**Formal analysis:** Abid Hussain, Kashif Kamran, Zafar Ullah, Shehla Shehla, Muhammad Kashif Obaid, Nazeer Ahmed, Qaiser Khan, Iram Liaqat.

**Investigation:** Asmatullah Kakar, Mahrukh Naseem, Nazeer Ahmed, Qaiser Khan, Iram Liaqat.

**Methodology:** Asmatullah Kakar, Kashif Kamran, Zafar Ullah, Shehla Shehla, Muhammad Kashif Obaid.

**Resources:** Abid Hussain, Shehla Shehla, Muhammad Kashif Obaid, Nazeer Ahmed, Qaiser Khan, Iram Liaqat.

**Supervision:** Asmatullah Kakar, Mahrukh Naseem.

**Validation:** Kashif Kamran, Zafar Ullah, Shehla Shehla, Muhammad Kashif Obaid.

**Writing – original draft:** Abid Hussain, Kashif Kamran, Zafar Ullah, Shehla Shehla, Muhammad Kashif Obaid.

**Writing – review & editing:** Asmatullah Kakar, Kashif Kamran, Zafar Ullah, Shehla Shehla, Muhammad Kashif Obaid.

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
